# A non-dispersion strategy for large-scale production of ultra-high concentration graphene slurries in water

Lei Dong[1,2], Zhongxin Chen [2,3], Xiaoxu Zhao [2,3], Jianhua Ma[1], Shan Lin[1], Mengxiong Li[1], Yang Bao [2], Leiqiang Chu[2], Kai Leng[2], Hongbin Lu[1] & Kian Ping Loh[2]

It is difficult to achieve high efficiency production of hydrophobic graphene by liquid phase exfoliation due to its poor dispersibility and the tendency of graphene sheets to undergo π–π stacking. Here, we report a water-phase, non-dispersion exfoliation method to produce highly crystalline graphene flakes, which can be stored in the form of a concentrated slurry (50 mg mL$^{-1}$) or filter cake for months without the risk of re-stacking. The as-exfoliated graphene slurry can be directly used for 3D printing, as well as fabricating conductive graphene aerogels and graphene–polymer composites, thus avoiding the use of copious quantities of organic solvents and lowering the manufacturing cost. This non-dispersion strategy paves the way for the cost-effective and environmentally friendly production of graphene-based materials.

[1] State Key Laboratory of Molecular Engineering of Polymers, Collaborative Innovation Center of Polymers and Polymer Composite, Materials and Department of Macromolecular Science, Fudan University, Shanghai 200433, China. [2] Department of Chemistry and Centre for Advanced 2D Materials (CA2DM), National University of Singapore, 3 Science Drive 3, Singapore 117543, Singapore. [3] NUS Graduate School for Integrative Sciences and Engineering, National University of Singapore, Centre for Life Sciences, #05-01, 28 Medical Drive, Singapore 117456, Singapore. Lei Dong and Zhongxin Chen contributed equally to this work. Correspondence and requests for materials should be addressed to H.L. (email: hongbinlu@fudan.edu.cn) or to K.P.L. (email: chmlohkp@nus.edu.sg)

Following the discovery of graphene, many optimistic predictions have been made with regards to its applications in fields ranging from electronics to medicine[1–4]. Due to the lack of efficient large-scale production processes, the production of graphene faces a dilemma where a compromise has to be made between scalability and graphene quality[4–14]. Liquid-phase exfoliation is a promising approach to realize scalable production of high-quality graphene or graphene oxide. Graphite can be exfoliated in organic or aqueous solutions containing a surfactant[15] by applying mechanical force (e.g., ultrasonic agitation[5], mechanical shearing[9], or ball-milling[11])[5,16]. However, large quantities of solvent is always needed in these processes for purification and dispersion due to the limited dispersion stability of graphene flakes. Only a very small amount of graphene, typically <1 mg mL$^{-1}$, can be dispersed in common solvents, and this is only marginally improved with the help of dispersing agents like superacids[17] or ionic liquids[18,19], or through sonication for extended periods of time[19,20]. Reducing the amount of solvent destabilizes the graphene dispersion and leads to the re-stacking of graphene by van der Waals interactions. This means that the production of 1 kg graphene by such dispersion approaches requires at least 1 ton of solvent, which is environmentally unfriendly and economically infeasible.

Three approaches have been explored previously to increase the concentration of graphene dispersions, which include the selection of a suitable solvent with a low enthalpy of mixing with graphene[15,16], the introduction of cation–π interaction between graphene and the solvent[18], and the creation of electrostatic repulsion between graphene flakes by protonation[17,21] or adding surfactants[22]. Generally, these approaches are only partially effective because of the large ratio of area to thickness (>10$^3$) and the tendency of graphene flakes to undergo π–π stacking[23]. Chemical intercalation-exfoliation methods have also been explored to prepare high-quality graphene in high yields[18,24,25]. However, these methods require the use of designer ionic liquids[18] and have limited scale-up capability (1 mg per batch)[24], and suffer from the drawbacks of producing partially exfoliated flakes (>10 nm in thickness)[25]. Rheological analysis of graphene dispersions indicates that the viscosity increases steeply with increasing graphene contents; for instance, it was observed that graphene-ionic liquid systems had a critical gel concentration as low as 4.2 mg mL$^{-1}$[26]. The increased viscosity will reduce the efficiency of liquid phase exfoliation and limit its large-scale production. Furthermore, the strong inclination for graphene to undergo π–π stacking has to be overcome and one way to do this is by introducing segregating agents.

In contrast to the conventional exfoliations in a solvent-dispersed system, here, we propose a non-dispersion strategy in which graphene is produced and stored as a flocculated aqueous slurry with concentrations as high as 50 mg mL$^{-1}$ (5 wt%). The presence of adsorbed ions prevents the re-stacking of graphene flakes and enables their re-dispersion in solution on demand. Partially oxidized graphite is used as the precursor, which is exfoliated by high-rate shearing in an alkaline aqueous solution of pH = 14. Our calculation of inter-sheet interaction energies indicates that under alkaline conditions, the ionization of oxygen-containing groups on graphene layers, even at very low concentrations (e.g., 5.9 atom%), can generate a large electrostatic repulsion energy ($E_E$) to counteract the interlayer van der Waals attraction energy ($E_{vdW}$). Due to the high ionic strength of the solution, exfoliated graphene flakes tend to form low-viscosity, flocculated slurry rather than a stable dispersion. Such graphene slurry can be easily re-dispersed in N-methyl-2-pyrrolidone (NMP) or alkaline water (pH = 12), and serves as a stock solution of graphene. Such graphene slurry possesses a three-dimensional (3D), loosely stacked microstructure with tunable modulus and viscosity, which can be directly used for 3D printing to form graphene aerogels and conductive polymer materials, without additional dispersion processes.

## Results

**Non-dispersion strategy for graphene production.** To reduce the use of solvent, a non-dispersion strategy was applied to mass produce graphene and fabricate functional materials directly using the flocculated slurry, as opposed to conventional liquid phase method where a large amount of solvent was used (Fig. 1).

Pristine graphite was partially oxidized using a very low molar ratio of oxidizer to carbon in graphite (0.076) to generate a low density of ionizable oxygen-containing groups on graphene layers. Partial oxidation-induced peak at 22.5° in the X-ray diffraction (XRD) spectrum indicates the formation of a stage-1 graphite intercalation compound with an interlayer distance of 8.0 Å (Supplementary Fig. 6)[27,28]. The pretreated graphite was then exfoliated by applying high-speed shear (Fig. 1b) in an alkaline aqueous solution (pH = 14). Owing to the high ionic strength, the exfoliated graphene flakes instantly flocculated to form a highly concentrated graphene slurry (5 wt% for solid

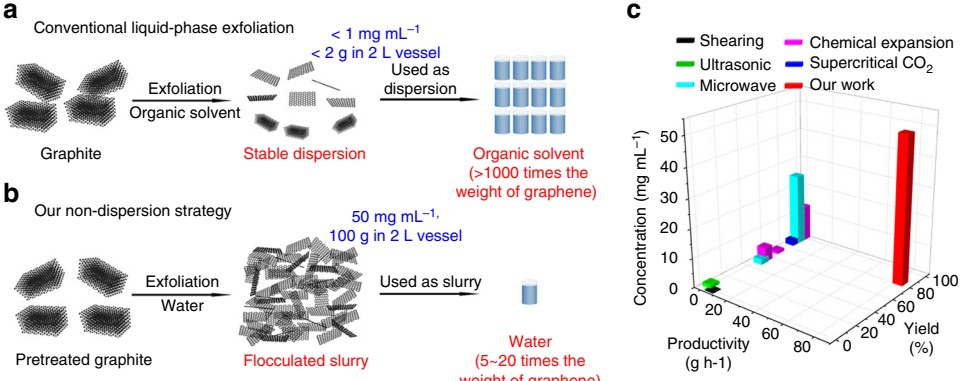

**Fig. 1** Preparation of ultra-high concentration graphene in water. **a** Conventional liquid-phase exfoliation where graphene flakes are peeled off from graphite and dispersed in an organic solvent, yielding low concentration dispersion in low yield because of the limited stability of the dispersion. **b** Our non-dispersion strategy for graphene production in water. Pretreated graphite is exfoliated by high-rate shearing and subsequently flocculated in alkaline water, producing graphene slurry on a large scale (100 g), high yield (82.5 wt%), and ultra-high concentration (50 mg mL$^{-1}$). **c** Comparison of concentration, yield, and production scale of non-dispersion strategy with other liquid-phase exfoliation strategies (shearing[9], ultrasonic[15,16,21], microwave[12,18], and chemical intercalation[5,24,25]-assisted exfoliation)

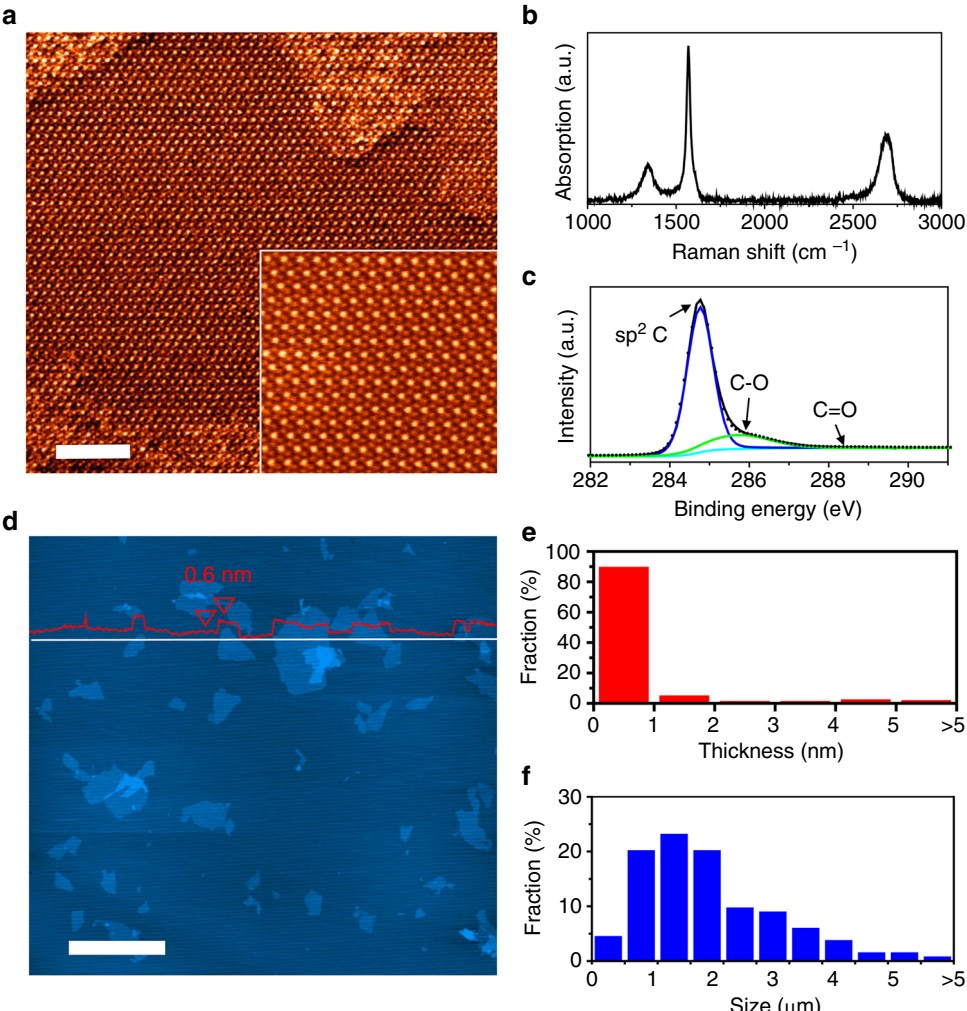

**Fig. 2** Quality of the exfoliated graphene flakes. **a** Atomic resolution STEM image of a graphene flake. Inset shows the corresponding magnified image with perfect graphene lattice. **b** Raman spectra of graphene, showing an $I_D/I_G$ ratio of 0.23. **c** XPS $C_{1s}$ spectrum of graphene. **d–f** Wide-field AFM image of graphene flakes and the corresponding thickness (**e**) and lateral size histograms (**f**). Scale bar: **a** 2 nm; **d** 5 μm

content, 2 L) within 2 h with a yield of 82.5 wt% (with respect to the weight of raw graphite) and a production efficiency of 82.5 g h$^{-1}$. The yield of highly concentrated graphene and efficiency of graphene production are superior to those in other liquid-phase exfoliation techniques in Fig. 1c. The slurry can be further concentrated up to 23 wt% of solid content by centrifugation or filtration. Such slurry can serve as stock solution of graphene, which can be re-dispersed in NMP or alkaline water (pH = 12) even after standing for over a week (Supplementary Fig. 7). XRD of the aged slurry shows that no π–π stacking occurred in Supplementary Fig. 6. This demonstrates the excellent stability of graphene slurry, an important prerequisite for storage, transportation, and application of graphene flakes.

Meanwhile, the viscosity of system is a critical factor in liquid-phase exfoliation but it is often overlooked[9–16]. Due to its large aspect ratio, graphene easily forms sediments or gels in solution when exceeding the concentration limit of stable graphene dispersion (<1 mg mL$^{-1}$)[15,16,21,22,26]. Although super acids or special ionic liquids may improve the dispersibility of graphene, these systems have high viscosity and limited exfoliation efficiency, and thus impractical for large-scale production[17,18]. In contrast, our graphene slurry displays a low shear viscosity of 0.064 Pa s at 50 s$^{-1}$ at 5 wt% solid content, which is over one order of magnitude lower than those of other dispersion systems

in Supplementary Fig. 17[29,30]. The viscosity at practical shear rate of 20,000 rpm, corresponding to 2094 s$^{-1}$, may even be lower due to shear thinning effect. This affords opportunities for high concentration exfoliation and production of high-quality graphene sheets.

**Characterization of graphene flakes**. The quality of graphene produced was assessed by scanning transmission electron microscopy (STEM), Raman spectroscopy, and X-ray photoelectron spectroscopy (XPS). In order to probe the quality of individual flakes, as-prepared graphene slurry was washed and re-dispersed prior to test. Atomic-resolution STEM image in Fig. 2a reveals the characteristic honeycomb lattice with long-range periodicity, confirming that the crystal structure of graphene is well retained after partial oxidation and shear-exfoliation. Raman spectra show two characteristic bands at 1325 cm$^{-1}$ (D band) and 1580 cm$^{-1}$ (G band), corresponding to the contributions from $sp^3$ type carbon from defects and $sp^2$ hybridized aromatic carbon in Fig. 2b. The $I_D/I_G$ mapping clearly evidences that most of the defects are located at the edges (Supplementary Fig. 5). The presence of the 2D band at ~2700 cm$^{-1}$ reflects the well-preserved aromatic structure of graphene, which is absent or negligible in reduced graphene oxide (rGO). The $I_{2D}/I_G$ ratio is ~0.45, corresponding to that of ~3 layer graphene[31]. Since we have to spin-

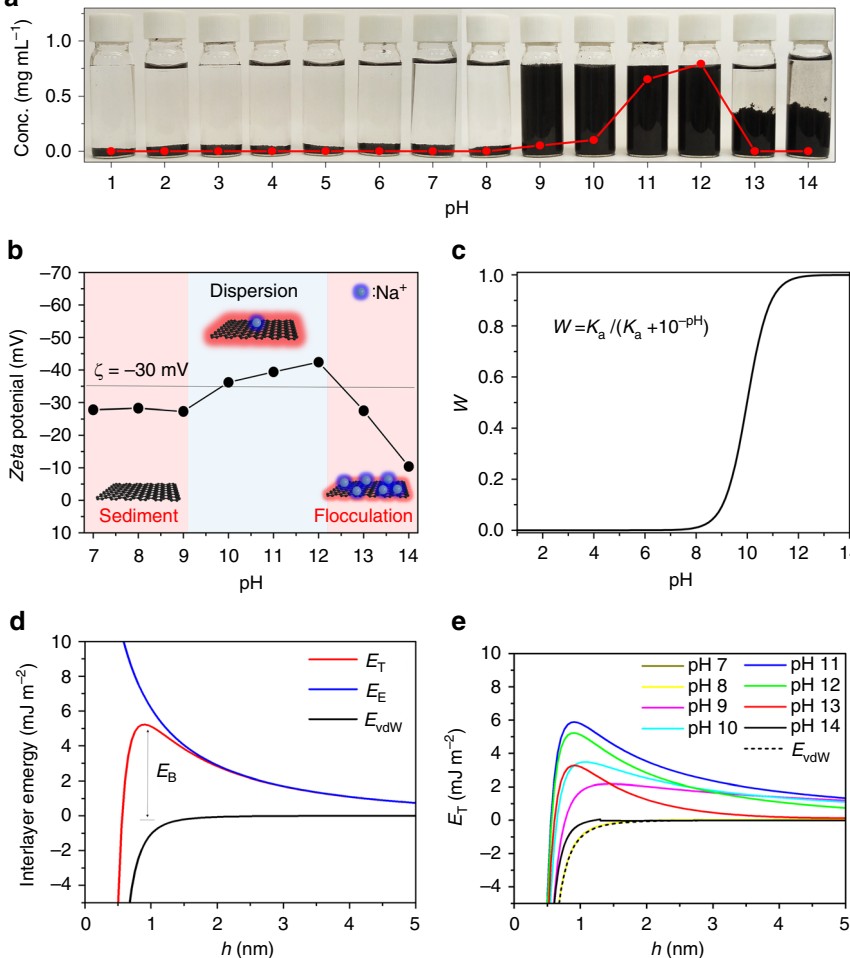

**Fig. 3** Stability of aqueous solution of our graphene with respect to pH. **a** Graphene aqueous solutions in the pH range of 1-14 and the corresponding maximum graphene concentrations (dotted line). **b** Zeta potential reveals the different dispersability of graphene solution as a function of pH. **c** Degree of ionization of phenolic hydroxyl groups on graphene flakes as a function of pH. **d** Interlayer interaction energies versus $h$ at pH = 12 from the DLVO theory. $E_T$ is given by the sum of $E_E$ and $E_{vdW}$, where the point denoted by $E_B$ in the $E_T$ versus $h$ curve determines the stability of the graphene solution. **e** $E_T$ versus $h$ curves at different pH values in the range of 7-14

coat NMP dispersion on Si wafer for Raman, it is difficult to avoid the re-stacking of graphene nanosheets during solvent evaporation. The exfoliated graphene only contains a slightly higher oxygen content than pristine graphite (5.9 versus 2.3 atom%) from XPS and elemental analysis in Supplementary Table 1. The corresponding $C_{1s}$ spectrum shows a strong C=C peak at 284.7 eV together with a small tail at 286.4 eV from C–O bonds, while the carbonyl species at 288.6 eV is negligible[25,32].

The morphology of graphene flakes was further analyzed by transmission electron microscopy (TEM), scanning electron microscopy (SEM) and atomic force microscopy (AFM). SEM and high-resolution TEM images show micron-sized, single-layer graphene flakes with the typical six-fold symmetry selected-area electron diffraction (SAED) pattern in Supplementary Figs. 1, 3 and 4. This is consistent with the AFM images of the graphene flakes (Fig. 2d and Supplementary Fig. 2), where a statistical analysis of over 100 flakes shows that >90 % of the flakes are single layer (<1 nm in thickness) with a lateral size ranging from 0.5 to 5 μm (Fig. 2e, f). As a result, graphene film obtained by vacuum filtration exhibits a highly hydrophobic surface with a water contact angle of 89.6°, which is comparable to pristine graphene[33]. The electrical conductivity is measured as $2.5 \times 10^4$ S m$^{-1}$ and can be further improved to $4.2 \times 10^4$ S m$^{-1}$ by HI reduction (Supplementary Fig. 9). Such conductivities are among

the best values for liquid phase exfoliated graphene, with the reference values tabulated in Supplementary Tables 2 and 3[9,15–18,24,26,34].

**Mechanism of non-dispersion exfoliation and aqueous dispersion.** It is challenging to obtain a stable graphene aqueous solution without adding surfactants[21]. The stability of our graphene aqueous dispersion is pH-dependent, with a maximum dispersion concentration at pH = 12 (Fig. 3a)[35]. Beyond pH = 12, the graphene dispersion becomes unstable due to ion-induced flocculation. Zeta potential (ζ) is a common indicator for the stability of nanomaterials. As shown in Fig. 3b, the graphene dispersion has a maximum ζ value of −42.4 mV at pH = 12, suggesting a strong electrostatic repulsive interaction between graphene flakes. This observation can be explained by the $C_{1s}$ XPS spectrum, where hydroxyl groups (−OH) are the primary oxygen-containing groups on our graphene. Since the dissociation constant of a phenolic hydroxyl group ($pK_a$) is ~10.0[36], the degree of ionization ($W$) increases at higher pH values (Fig. 3c). At pH values higher than 12, the hydroxyl groups on graphene will be fully ionized. Nevertheless, it remains an open question whether the presence of 5.9 atom% of oxygen-containing groups is sufficient to generate an electrostatic repulsive force against π–π re-stacking.

To investigate the interactive forces governing the dispersion of our graphene, we employ the classical Derjaguin–Landau–Verwey–Overbeek (DLVO) theory to explain our experimental observations. Two adjacent graphene flakes in the dispersion are treated as two parallel plates with a separation distance of $h$, where the electrical double layer (EDL, thickness denoted by $\kappa^{-1}$) due to negatively charged oxygen groups on the surface and adsorbed counter ions determines the dispersion stability (Supplementary Fig. 14). The total interaction energy ($E_T$) is given by the sum of the electrostatic repulsion energy $E_E$ and the van der Waals attraction energy $E_{vdW}$[37] or, $E_T = E_E + E_{vdW}$. For graphene flakes, $E_{vdW}$ is given by $E_{vdW} = -d_0^4\gamma/h^4$[22], where $d_0 = 0.335$ nm and $\gamma$ is the surface energy (70 mJ m$^{-2}$). $E_E$ depends on the electrostatic potential on the graphene surface ($\psi_0$), the electrolyte concentration ($c$), and the flake separation ($h$)[38,39]. In a previous study, $E_E$ was calculated by measuring the value of *Zeta* potential ($\zeta$), which gives the change of $E_E$ as a function of $h$[22]. However, this calculation is applicable only when $\zeta < 25$ mV[22]. To overcome this limit, we have calculated $E_E$ from available data for $\psi_0$, $h$, and other constants (Supplementary Eqs. (2) and (3))[38,39]. Using the Poisson–Boltzmann (PB) equation to derive the relationship between $\psi_0$ and surface charge density ($\sigma$), we can write $\sigma = 4Lce\sinh(e\psi_0/2kT)/\kappa$[39], where $L$ is the Avogadro constant, $e$ is the elementary electronic charge, $k$ is the Boltzmann constant, $\kappa$ is the Debye–Hückel parameter, and $T$ is the absolute temperature. Assuming that all oxygen-containing groups on our graphene sheets are hydroxyl group, and $\sigma$ is dependent on the density of −OH groups together with the degree of ionization $W$ at different pH values, $\sigma$ can be also expressed as $\sigma = WAe/(2.62\times10^{-20})$ (see Supplementary Notes 1 and 2), where

$A$ is the molar ratio of oxygen atoms to carbon atoms, that is:

$$\sigma = W \times \frac{Ae}{2.62\times10^{-20}} = \frac{4Lce}{\kappa}\sinh\left(\frac{e\psi_0}{2kT}\right) \qquad (1)$$

At pH = 12, the obtained $E_E$ curve for graphene dispersion is presented in Fig. 3d, revealing that the repulsive force increases with decreasing $h$. After taking $E_{vdW}$ into account, the variation of the total interaction energy $E_T$ with $h$ between two adjacent graphene flakes is displayed in Fig. 3d. A large energy barrier ($E_B = 5.22$ mJ m$^{-2}$) against aggregation is observed, which is ~8 times larger than that of a surfactant-stabilized graphene dispersion (~0.64 mJ m$^{-2}$)[22]. Even at oxygen coverage of 1 atom %, the calculation result also shows a relatively large energy barrier (1.83 mJ m$^{-2}$) in Supplementary Fig. 16 to prevent re-stacking. Figure 3e and Supplementary Fig. 15 further provide the $E_E$ and $E_T$ curves at different pH values. $E_E$ is nearly zero in the pH range of 7~8 due to the lack of ionized oxygen-containing groups, leading to the re-stacking of graphene flakes. In contrast, a large number of counter-ions (Na$^+$) will be adsorbed onto the surface of graphene when pH > 13. This results in the compression of the EDL from 3.0 nm at pH = 12 to 0.3 nm at pH = 14 (Supplementary Table 4), and the flocculation of graphene flakes. We observe a similar flocculation behavior when an equivalent amount of NaCl was added to the graphene dispersion at pH = 12, suggesting that the absorbed ions prevent the π−π aggregation of graphene flakes[40], similar to the flocculated clay nanoplatelets[37,41] (Supplementary Fig. 12).

The mechanism of non-dispersion exfoliation can be explained by the pH-dependent stability of graphene dispersions. Indeed,

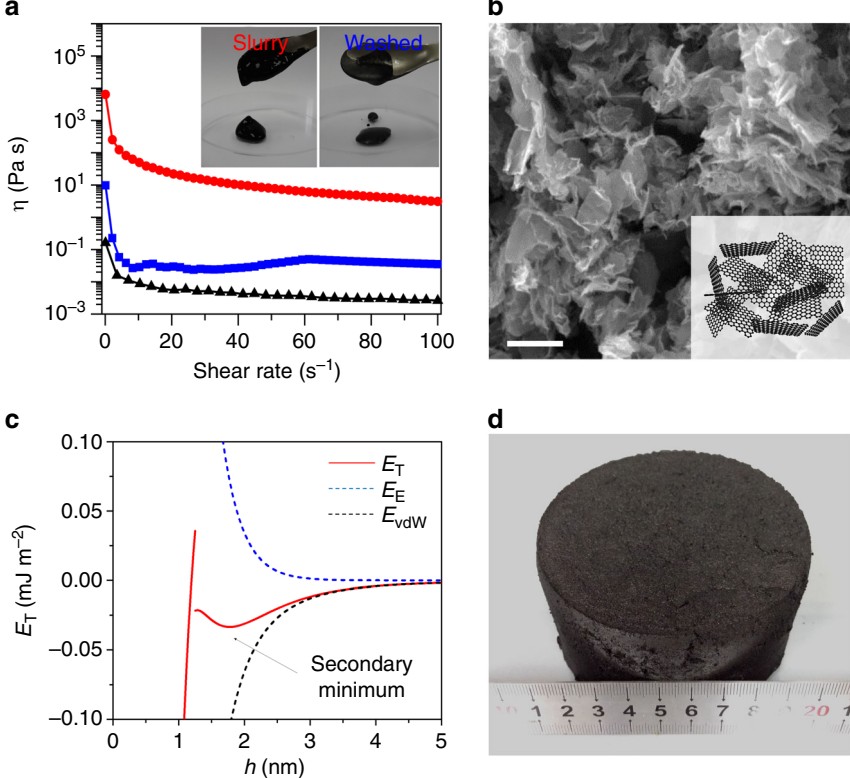

**Fig. 4** Structure and dispersibility of graphene slurry. **a** At the same concentration of 8.1 wt%, graphene slurry (red dotted line) exhibits significantly higher shear viscosity than the washed counterpart (blue dotted line) and raw graphite suspension (black dotted line). Inset shows the corresponding photos. **b** SEM image of graphene slurry with loosely stacking structure. Inset shows the corresponding structure model. **c** Enlarged $E_T$ versus $h$ curve of graphene slurry at pH = 14, highlighting the energy trough, called secondary minimum at a distance of 1.8 nm. **d** Graphene slurry was filtered through a G4 filter funnel, obtaining a wet graphene cake with a solid content of 23 wt%. Scale bar: **b** 20 μm

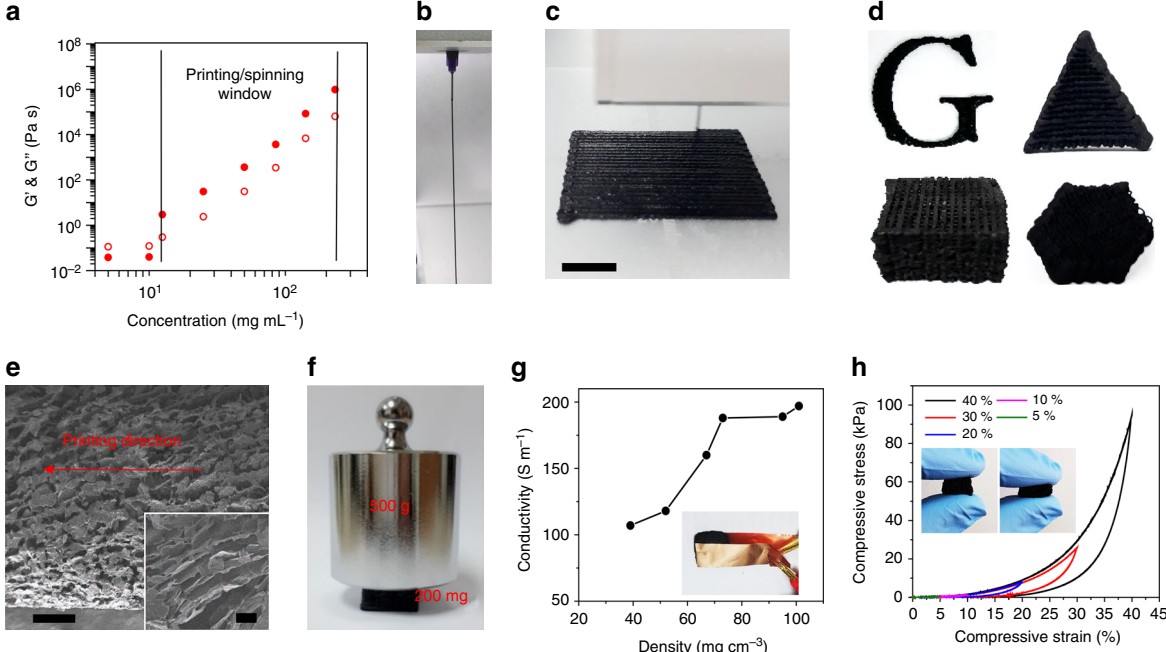

**Fig. 5** Fabrication of graphene composites by 3D printing of concentrated slurry. **a** Log−log plots of the modulus (solid dots for storage modulus and open circles for loss modulus) at 1 Hz versus the concentration of graphene slurry. The marked concentration window where storage modulus is over loss modulus provides a suitable window for the direct processing of printing and spinning. **b**, **c** 3D printing of graphene slurry at 10 wt%. **d** 3D-printed graphene architectures with variable shapes. **e** Microstructure of graphene fiber in the printed aerogel. **f** Image of the printed graphene aerogel (200 mg, 50 mg cm$^{-3}$) supporting 2500 times of its weight. **g** Electrical conductivity of the printed aerogel versus density. **h** Stress−strain curve during loading−unloading cycles by increasing strain amplitude for PDMS−graphene composites. Scale bar: **c** 1 mm; **e** 200 μm; inset of **e** 10 μm

the yield of graphene is very low (< 1.2 wt%) when the pretreated graphite is exfoliated at pH = 12, even though the dispersion stability is highest at this pH value. This is attributed to the inherent low dispersibility of graphene which sets a limit on the yield of high-concentration exfoliation. In contrast, when exfoliated at pH = 13 or 14, graphene flakes rapidly flocculate due to the compressed EDL, which reduces the system viscosity and facilitates exfoliation. The flocculated graphene flakes are unable to form π−π stacking due to the presence of absorbed ions and the loose stacking, so that they can be re-dispersed in NMP or alkaline solutions.

**Storage and dispersion of graphene slurry**. The microstructure of graphene slurry largely affects its solution processing capability and applications. Actually, we found that the morphology and solution behavior of graphene slurries are also pH-dependent. As shown in Fig. 4a and Supplementary Fig. 18, as-obtained graphene slurry (pH = 14) behaves like an elastic gel which has a steady viscosity over two orders of magnitude greater than that of the washed graphene slurry at pH = 7. The latter exhibits a fluid-like behavior, similar to the graphite suspension, for which the viscosity is controlled solely by the weak interaction between water molecules and graphite plates. To observe their microstructure, two samples were prepared through liquid nitrogen quenching and freeze-drying to avoid possible structural changes. As shown in Fig. 4b, as-obtained graphene slurry has a loosely stacking structure, similar to that of the flocculated inorganic clays[37,41]. In contrast, the slurry at pH = 7 exhibits a face-to-face aggregation due to the lack of electrostatic repulsion from charged functional groups (Supplementary Fig. 19a). In addition, we observed a ~3-fold volume shrinkage and one order of magnitude decrease in the specific surface area (SSA) from pH = 14 to pH = 7, which also reflects pH-dependent structure changes in graphene slurries (Supplementary Fig. 19).

To further quantify the stability of two graphene slurries, we calculate the interlayer interaction energies for the graphene slurries at pH = 14 and 7, respectively. As shown in Fig. 4c, a trough (the secondary minimum) appears at h = 1.8 nm at pH = 14 because the electrostatic energy $E_E$ decreases more rapidly with increasing distance compared to $E_{vdW}$[42]. Moreover, since this trough is quite shallow (0.033 mJ/m$^2$), the flocculated graphene slurry is re-dispersible, even in form of graphene cake with extreme high loading (23 wt%, Fig. 4d)[37]. For the case at pH = 7, however, $E_T$ decreases monotonically with h and has a deep primary minimum appeared at the very small h, where the van der Waals attractive force predominates, leading to unfavorable π−π aggregations[37].

**3D printing of concentrated graphene slurry**. High-concentration graphene slurries are highly desired for the fabrication of many functional materials, for instance, printing or spin-coating typically requires a work window of high solid contents in Fig. 5a. The log−log plot of graphene content (c) against storage/loss modulus reveals a critical gel concentration ($c_g$) of ~1.25 wt% for the graphene slurry at pH = 14. Below this $c_g$, the flocculated graphene flakes form individual micro-scale flocs. The system exhibits a liquid-like behavior for which loss modulus (G″) is greater than storage modulus (G′) in the measured frequency range. Beyond $c_g$, where G″ is larger than G″, the slurry behaves like an elastic gel. Here, we have fabricated various graphene aerogel structures via 3D-printing of highly concentrated slurry at the elastic gel region in Fig. 5d. To the best of our knowledge, this is the first demonstration of water-phase 3D-printing of exfoliated graphene[43,44]. Previous efforts on graphene aerogels rely on the sol-gel chemistry, which is challenging for large-scale production[45,46]. The printed aerogel is macroporous (Fig. 5e, f) with good mechanical strength. Its SSA is determined

from methyl blue (MB) absorption measurements ($1240 \, m^2 \, g^{-1}$, Supplementary Fig. 19b). The electrical conductivity reaches ~197 S m$^{-1}$ at a density of 100 mg cm$^{-3}$ (Fig. 5g), which is comparable to 3D-printed rGO networks by conventional dispersion approaches[43,44], although it is inferior to that of CVD-grown method[47]. By incorporating high-temperature annealing in commercial graphite production, it is possible to further improve the electrical conductivity to a level comparable with that of CVD-grown aerogels[48]. The printed graphene aerogels can be used as 3D templates for in-situ polymerization, which are promising for applications in a wide range of energy storage devices[49] and durable absorbent materials[50]. Owing to the porous structure of the graphene aerogels, the monomer can diffuse rapidly into the graphene framework. After polymerization, polydimethylsiloxane (PDMS)–graphene composites were obtained, where PDMS was evenly distributed in the graphene framework. The resulting composite exhibits nonlinear super-elastic behavior and ultra-large, reversible compressibility with a strain up to 40%. Multi-cycle compression test also shows that after the first loading–unloading loop, the stress was still stabilized at 53 kPa in the following nine loops (Supplementary Fig. 20), demonstrating a stable, bi-continuous texture in the PDMS–graphene composites.

In conclusion, we have demonstrated an industrially viable water-phase exfoliation strategy for preparing high-quality graphene and composites. This approach bypasses the destructive chemical oxidation processes, avoids the use of copious quantities of solvent, and addresses the critical issues related to the storage and transportation of graphene. Using this strategy, graphene flakes can be exfoliated in the form of highly concentrated slurries (5 wt%) with high production efficiencies (82~170 g h$^{-1}$). The exfoliated flakes form loosely stacked, flocculated aggregates due to the presence of adsorbed ions on the weakly oxidized surface. Such graphene slurries possess a 3D loosely stacking microstructure with rheological properties that are markedly different from that of closely stacked graphene flakes; for example, they can be directly 3D-printed to fabricate conductive graphene aerogels and be used to fabricate high graphene content composites. Different from traditional oxidation-reduction approaches, this non-dispersion exfoliation strategy allows a cost-effective, large-scale production, storage and transport of graphene in aqueous medium.

## Methods

**Pretreatment of graphite**. Pretreated graphite was obtained using the conventional intercalation process using sulfuric acid. KMnO$_4$ (100 g, 1 wt equiv.) was added in batches into concentrated H$_2$SO$_4$ (2 L, 98%) over a period of 30 min in an ice-water bath. Then the ice-water bath was removed and natural graphite flake (100 g, 1 wt equiv., 500 µm) was added. The system was stirred at 35 °C for 2 h. After reaction, the black flakes were filtered through a 200-mesh sieve (Supplementary Fig. 11 and Movie 1) and separated from dark green solution. Concentrated H$_2$SO$_4$ was recovered for recycling. Then the filter cake was poured into 2 L of ice water to avoid sudden increase in temperature. Fifty milliliters of 30 wt% H$_2$O$_2$ was added to decompose the insoluble manganese dioxide. After filtering and washing, wet powders of pretreated graphite were obtained for subsequent water-phase exfoliation.

**Water-phase exfoliation**. Typically, 100 g of pretreated graphite (based on the weight of raw graphite) was added into 2 L of 1 M NaOH aqueous solution (pH = 14 alkaline water by adding 80 g of NaOH into 2 L of DI water). The mixture was subjected to shear at 20,000 rpm for 1 h at r.t. by using an FA 40 high shear dispersing emulsifier (Fluko) with a working tool, resulting in a black graphene slurry. For yield calculation, the graphene slurry was centrifuged at 10,000 rpm for 10 min and repeatedly washed by a large amount of water (4~6 times) until pH approaches 10. Then the washed slurry was re-dispersed in NMP at a concentration of ~0.1 mg mL$^{-1}$ and sonicated for 1 min. The resulting graphene–NMP dispersion was centrifuged at 2000 rpm for 30 min. The supernatant in the upper two-third solution was collected and the precipitate was re-dispersed in NMP. The centrifugation and re-dispersing processes were repeated until the supernatant is colorless. The collected upper and lower sections of the supernatant were dried

separately in a vacuum oven at 60 °C for 10 h. Finally, the mass yield of graphene flakes in slurry was calculated by the mass ratio of dried upper graphene to raw graphite. Control experiments can be found in Supplementary Figs. 8, 12 and 13. Large-scale production of 1 kg graphene is provided in Supplementary Fig. 10.

**3D printing of graphene slurry**. As-exfoliated graphene slurry at pH = 14 was concentrated to a suitable concentration (from 8 to 23 wt%) by filtration through a G4 funnel and subsequently used as a printable ink without modification. The concentrated graphene slurry was transferred to a 20 mL syringe barrels and printed using a robotic deposition device. The diameter of the printing nozzles ranged from 200 to 600 µm. The pressure was regulated during printing in air to maintain constant ink flow. After printing, the printed structures were frozen at −20 °C for 5 h and freeze-dried for 24 h. Then the dried structures were carefully immersed into water to remove the residual ions. Finally, graphene aerogels was obtained by freeze-drying process for 24 h.

**Equipment**. The following equipment was used: High-rate shearing (FA 40 emulsifying machine, Fluko), STEM (JEOL JEM-ARM200F with aberration-correction, 60 kV), TEM/SAED (Tecnai G$^2$ 20 TWIN and JEM-2100F, 200 kV), AFM (Multimode 8), XPS (AXIS UltraDLD, monochromatic Al $K_\alpha$), Rheology (HAAKE MARS III, cone-plate or cylinder geometry at 25 °C), SEM (Ultra 55), Raman (XploRa, 532 nm), XRD (PANalytical X'Pert PRO, Cu $K_\alpha$, operated at 40 kV and 40 mA), UV-Vis (Lambda 35), *Zeta* (ZS90), EA (vario EL III), Conductivity (SX1944, four-point probe), Contact angle (JC2000 DM), 3D-printing (Bio-Architect®–Pro), Multi-cycle compression (Reger-RWT10).

**Data availability**. All data are available from the authors upon reasonable request.

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

## Acknowledgements

H. Lu appreciates financial support from 973 project (2011CB605702), the National Science Foundation of China (51173027), and Shanghai key basic research project (14JC1400600). K.P. Loh thanks National Research Foundation, Prime's Minister Office for support for NRF Investigator award "Graphene oxide a new class of catalytic, ionic and molecular sieving materials, award number NRF-NRF12015-01". L.D. acknowledges financial support from Fudan University. Z.C. acknowledges financial support in the form of a NGS scholarship.

## Author contributions

L.D. and Z.C. conceived the research and wrote the draft. L.D. conducted the exfoliation of graphene and DLVO calculations. X.Z. and Z.C. conducted TEM characterization and data analysis. L.D. and J.M. performed the 3D printing and performance analysis. Z.C., S. L., Y.B., L.C., J.M. and K.L. assisted in sample preparation and materials characterization. H.L. and K.P.L. supervised the research. All authors discussed and commented on this manuscript.

## Additional information

**Competing interests:** The authors declare no competing financial interests.

