## [Peer Review File · Nature Communications]

Reviewers' comments:

Reviewer #1 (Remarks to the Author):

Review of NCOMMS-17-20613

This manuscript presents a novel method for preparing high concentration slurries of exfoliated graphene flakes. The method is especially important given the many graphene-based technologies that would benefit for a large-scale, economical source of high quality graphene. For this reason, I recommend publication of this manuscript with the following revisions.

1. Authors state that pi-pi stacking is irreversible. (line 54) This is not true and should be corrected. Pi-pi stacking is by its very nature reversible.
2. What technique/instrument was used to apply high speed shear? (Line 81-82) Please include relevant settings/parameters.
3. How was the alkaline solution prepared? (line 82) What specific base was used? Concentrations? More details are needed.
4. Please measure and report the BET surface area of the 3D printed aerogels.
5. Authors should cite the following existing work and discuss the results in light of this manuscript's findings.
Graphene aerogels: JOURNAL OF THE AMERICAN CHEMICAL SOCIETY Volume: 132 Issue: 40 Pages: 14067-14069 Published: OCT 13 2010; ACS NANO Volume: 4 Issue: 7 Pages: 4324-4330 Published: JUL 2010 ;
Electrical conductivity: ACS NANO Volume: 8 Issue: 10 Pages: 11013-11022 Published: OCT 2014; Polymer-graphene composite: JOURNAL OF MATERIALS CHEMISTRY A Volume: 1 Issue: 10 Pages: 3495-3502 Published: 2013; JOURNAL OF MATERIALS CHEMISTRY A Volume: 2 Issue: 9 Pages: 3057-3064 Published: 2014

Reviewer #2 (Remarks to the Author):

The manuscript claims the processing graphene using an aqueous system for processing well dispersed graphene sheets up to 23 wt%. The processing method utilized high speed shearing of pre-treated graphite powders to exfoliate the sheets. Jamming interactions between graphene sheets prevents the sheets from aggregating during exfoliation at pH = 14. The pre-treatment of graphite involves exposing graphite to KMnO₄ and concentrated H₂SO₄, which partially oxidizes graphite layers. While partial oxidation generate enough ionic repulsion to improve exfoliation and stability, the π-π conjugated graphene structure remains intact for electronic conduction. As a demonstration for the slurry's application, the slurry was applied as a 3D printable gel, which can be further freeze-dried into porous scaffold. Overall, this is an excellent piece of work.

1. The article demonstrated a significant improvement to solution exfoliation of graphene. By partially oxidizing graphite particles, liquid exfoliation of graphene can be scaled up without significantly sacrificing graphene quality. This is a much sought after improvement to the preparation of graphene.

2, Line 60, the authors refer to pre-treated graphite as "sulfate-intercalated graphite". However, the graphene sheets' pH response and elemental analysis indicate that the sheets were partially oxidized. The pre-treatment method is also similar to graphene oxide synthesis with variation in molar ratios. Wouldn't it be better to call it "partially oxidized graphite"

3. Similarly for lines 79-81, partial oxidation would yield a similar shift in XRD spectrum.

4. The reported graphene slurry has a high concentration of flocculating single-layer graphene sheets at pH of 14, and authors compared this mixture to GO solutions with well dispersed nano-sheets. A more direct comparison is between GO solution and graphene slurry at pH = 12, the stable pH for storing and utilizing graphene slurry.

5. At line 253, the authors claim that sulfuric acid can be recovered and recycled. This may be easier said than done. Can the authors demonstrate this or provide relevant references to support this?

Reviewer #3 (Remarks to the Author):

The manuscript by Dong et al. proposes the new approach for the liquid exfoliation of graphite to yield graphene. To achieve better exfoliation, the authors subject graphite to partial oxidation with potassium permanganate is sulfuric acid, and then subject the as-obtained product to shear in aqueous solution of NaOH at pH=14. The difference from the previous studies is that the product is obtained not as stable dispersion, but in the form of flocculated aqueous slurry with concentrations as high as 5 wt%. The presence of adsorbed ions prevents the irreversible restacking of graphene flakes and enables their re-dispersion in solution on demand. The authors further demonstrate the use of the as-made graphene slurry for 3D printing. This approach is novel, and has great potential for practical applications, since avoiding copious amounts of water and/or organic solvents helps one to lower the manufacturing cost, and to minimize expenses with further storage and processing.

The only concern I have relates to the experimental data, confirming the quality of as-obtained graphene.

1. On the first step authors use 1:1 weight ratio of graphite-to-KMnO₄. At this ratio, significant oxidation of graphite is normally registered. See for example: Carbon 1991, 29, 469-474; Carbon 1995, 33, 1585-1592; Chem. Mater. 2006, 18, 2740-2749; Carbon 2013, 53, 38-49; ACS Nano, 2014, 8, 3060-3068. As shown in the latter study, at this graphite/oxidizer ratio, roughly one half of the body of a graphite flake is converted to graphite oxide. The question: what authors do differently, that they obtain almost non-oxidized graphene? Shorter exposure time, lower temperature, etc.? Explanation for this apparent discrepancy with the literature data should be given.

2. The Raman mapping (Fig. 2b) does not match the rest of the provided experimental data. The AFM and TEM images show single-layer graphene. The Raman spectrum does not represent the single-layer graphene flake. This is apparent from the G/2D ratio and the character of the D and G-bands. The Raman spectrum of a single-layered graphene is very different: 2D signal must be higher than G-band. Here the situation is opposite, strongly attributing the given spectrum to multi-layered graphene (or graphite) with number of layers >5. Both D-band and G-band are broadened, suggesting highly damaged structure, with the density of defects similar to that in GO

(this is logical in light of the oxidative treatment). Thus, the Raman spectrum represents the multi-layered graphene flake, where the top layer is fully oxidized, i.e. the top layer is graphene oxide. It is very much the same GNP as the product obtained in ref. [25], authors citing.

The question: if the authors obtain mostly single-layered graphene, as they show on the AFM images (Fig. 2,e; Fig. S2; Fig. S3), then why they picked the multi-layered oxidized flake for the Raman mapping? All the data should be in accordance with each other.

3. Based on Fig. 2b, I would expect the flakes to be thicker along the perimeter. The AFM height profiles for GO normally show 0.9-1.2 nm height. Graphene - 0.5-0.6 nm. Could the authors, please, comment on this in the manuscript text? If the flakes are indeed thicker on the edges, as I can see from the provided height-profiles for some flakes, discussing this in the text will strengthen the publication. I would recommend to acquire a high resolution image of one single flake and carefully examine it.

4. The SEM images of as-made graphene (Fig. 4b and Fig. S1c) also look more like those for multi-layered GNP. Thus, the two data (Raman and SEM) provided by the authors, suggest that they obtain GNP, but not single-layered graphene, as they claim. The 3D printing and the fabricated aerogels do not confirm the single-layer character.

The complete exfoliation of graphite source to the single-layer graphene is the main claim of this study that makes it, according to the authors, advantageous over the previous works. Thus, this needs to be unambiguously confirmed.

I will be happy to support after addressing these questions. Major revision is suggested.

Point-to-Point Response for

A Non-Dispersion Strategy for Large-Scale Production of Ultra-High Concentration Graphene Slurries in Water (NCOMMS-17-20613)

Reviewer #1:

*This manuscript presents a novel method for preparing high concentration slurries of exfoliated graphene flakes. The method is **especially important** given the many graphene-based technologies that would benefit for a large-scale, economical source of high quality graphene. For this reason, I recommend publication of this manuscript with the following revisions.*

Response: We thank the reviewer for his/her positive comments on our paper. We have addressed his/her concerns on the details of experimental setup, the BET surface area of 3D aerogels and the improper descriptions on pi-pi stacking in this revised version.

Question 1: *Authors state that pi-pi stacking is irreversible. (line 54) This is not true and should be corrected. Pi-pi stacking is by its very nature reversible.*

Response: We have deleted the improper descriptions on “irreversible” π - π stacking in line 54. Similar mistakes have been corrected throughout this manuscript.

Line 46: “the tendency of graphene flakes to undergo ~~irreversible~~ π - π stacking²³.”

Line 54: “Furthermore, the strong inclination for graphene to undergo ~~irreversible~~ π - π stacking has to be overcome and one way to do this is by introducing segregating agents.”

Line 59: “The presence of adsorbed ions prevents the ~~irreversible~~ re-stacking of graphene flakes and enables their re-dispersion in solution on demand.”

Line 208: “where the van der Waals attractive force predominates, leading to ~~irreversible-stacking~~ unfavorable π - π aggregations³⁷.”

Question 2: *What technique/instrument was used to apply high speed shear? (Line 81-82) Please include relevant settings/parameters.*

Response: We have included a more detailed description about the instrument used in the revised manuscript and the corresponding photo in the Supplementary Information, see below:

Line 259: “The mixture was subjected to shear at 20,000 rpm for 1 h at r.t. by using an FA 40 high shear dispersing emulsifier (Fluko) with a working tool, resulting in a black graphene slurry.”

Figure 1: Exfoliation process of 100 g graphite in an alkaline aqueous solution (2 L, pH = 14) by high speed shearing with an FA40 high shear dispersing emulsifier (Fluko).

Question 3: *How was the alkaline solution prepared? (line 82) What specific base was used? Concentrations? More details are needed.*

Response: We have included a more detailed description as follow:

Line 258: "...was added into 2L of 1M NaOH aqueous solution (pH = 14 alkaline water by adding 80 g of NaOH into 2 L of DI water)."

Question 4: *Please measure and report the BET surface area of the 3D printed aerogels.*

Response: We have supplied the specific surface area (SSA) measurements on 3D aerogels and corrected the wrong description on "microporous" in the revised version. Since our aerogels contain macropores and voids as shown in SEM images, we employ two methods, BET and liquid-phase methyl blue (MB) absorption, to measure the SSA value. However, the SSA value from BET ($243 \text{ m}^2 \text{ g}^{-1}$) is much lower than from MB absorption ($1240 \text{ m}^2 \text{ g}^{-1}$), indicating possible re-stacking of graphene sheets during drying process. Details on the difference between these two methods can be found in our previous publication (Room-temperature intercalation and ~1000-fold chemical expansion for scalable preparation of high-quality graphene. *Chem. Mater.* **2016**, 28, 2138-2146).

Line 220: "The printed aerogel is ~~microporous~~ macroporous (Fig. 5e-f) with good mechanical strength. Its specific surface area is determined from methyl blue (MB) absorption measurements ($1,240 \text{ m}^2 \text{ g}^{-1}$, Supplementary Fig. 19 b)."

Supplementary Figure 19. (b,c) Methyl blue (MB) absorption results of graphene slurries at pH = 14 and pH = 7. The adsorption quantities of MB at pH = 14 from UV-Vis spectra is 11.3 times higher than that of at pH = 7, corresponding to a specific surface area of $1240 \text{ m}^2 \text{ g}^{-1}$.

Question 4: *Authors should cite the following existing work and discuss the results in light of this manuscript's findings. Graphene aerogels: JOURNAL OF THE AMERICAN CHEMICAL SOCIETY Volume: 132 Issue: 40 Pages: 14067-14069 Published: OCT 13 2010; ACS NANO Volume: 4 Issue: 7 Pages: 4324-4330 Published: JUL 2010 ; Electrical conductivity: ACS NANO Volume: 8 Issue: 10 Pages: 11013-11022 Published: OCT 2014; Polymer-graphene composite: JOURNAL OF MATERIALS CHEMISTRY A Volume: 1 Issue: 10 Pages: 3495-3502 Published: 2013; JOURNAL OF MATERIALS CHEMISTRY A Volume: 2 Issue: 9 Pages: 3057-3064 Published: 2014*

Response: We have cited and commented on the above references in the revised version.

Line 220: “To the best of our knowledge, this is the first demonstration of water-phase 3D-printing of exfoliated graphene^{43,44}. Previous efforts on graphene aerogels rely on the sol-gel chemistry, which is challenging for large-scale production^{45,46}.”

Line 221: “The electrical conductivity reaches $\sim 197 \text{ S m}^{-1}$ at a density of 100 mg cm^{-3} (Fig. 5g), which is comparable to 3D-printed rGO networks by conventional dispersion approaches^{42,43} although it is inferior to that of CVD-grown method⁴⁴. By incorporating high-temperature annealing in commercial graphite production, it is possible to further improve the electrical conductivity to a level comparable with that of CVD-grown aerogels⁴⁸.”

Line 224: “The printed graphene aerogels can be used as 3D templates for in-situ polymerization, which are promising for applications in a wide range of energy storage devices⁴⁹ and durable absorbent materials⁵⁰.”

Reviewer #2:

The manuscript claims the processing graphene using an aqueous system for processing well dispersed graphene sheets up to 23 wt%. The processing method utilized high speed shearing of pre-treated graphite powders to exfoliate the sheets. Jamming interactions between graphene sheets prevents the sheets from aggregating during exfoliation at $\text{pH} = 14$. The pre-treatment of graphite involves exposing graphite to KMnO_4 and concentrated H_2SO_4 , which partially oxidizes graphite layers. While partial oxidation generate enough ionic repulsion to improve exfoliation and stability, the π - π conjugated graphene structure remains intact for electronic conduction. As a demonstration for the slurry's application, the slurry was applied as a 3D printable gel, which can be further freeze-dried into porous scaffold. Overall, this is an excellent piece of work.

Question 1: The article demonstrated a significant improvement to solution exfoliation of graphene. By partially oxidizing graphite particles, liquid exfoliation of graphene can be scaled up without significantly sacrificing graphene quality. This is a much sought after improvement to the preparation of graphene.

Response: We thank the reviewer for his/her positive comments on our paper. We have addressed his/her concerns on the partially oxidized graphite, control experiments on GO solution and the recovery of sulfuric acids in this revision.

Question 2: Line 60, the authors refer to pre-treated graphite as “sulfate-intercalated graphite”. However, the graphene sheets' pH response and elemental analysis indicate that the sheets were partially oxidized. The pre-treatment method is also similar to graphene oxide synthesis with variation in molar ratios. Wouldn't it be better to call it “partially oxidized graphite”

Response: We agree that “sulfate-intercalated graphite” is confusing since we employed a small amount of oxidizing agent (KMnO_4) during intercalation. It is known that oxidizing agents, even when used in a low molar ratio (1 wt. equiv. in this manuscript) and low temperature, can result in a certain degree of oxidation on graphite flakes (5.9 atom% O) (The chemistry of graphene oxide, *Chem. Soc. Rev.*, **2010**, 39, 228-240; Chemistry with graphene and graphene oxide - challenges for synthetic chemists, *Angew. Chem. Int. Edit.*, **2014**, 53, 7720-7738). In this regards, we have modified the “sulfate-intercalated graphite” into “partially oxidized graphite” throughout this manuscript.

Line 60: “~~Sulfate-intercalated~~ Partially oxidized graphite is used as the precursor, which is exfoliated by high-rate shear in an alkaline aqueous solution of $\text{pH} = 14$.”

Line 77: “Pristine graphite was *partially oxidized* using a very low molar ratio of oxidizer to carbon in graphite (0.076) in order to generate a low density of ionizable oxygen-containing groups on graphene layers.”

Line 107: “...confirming that the crystal structure of graphene is well retained after *sulfate-intercalation partial oxidation* and shear-exfoliation.”

Line 248: “Pretreated graphite was obtained using the conventional *intercalation process using sulfuric acid*.”

Question 3: *Similarly for lines 79-81, partial oxidation would yield a similar shift in XRD spectrum.*

Response: We agree that partial oxidation would yield a similar shift in XRD spectrum. Partial oxidation and graphite intercalation may proceed concurrently during the oxidation-intercalation of graphite. We have modified the term of “sulfate-intercalated graphite” into more accurate “partially oxidized graphite” throughout this manuscript.

Line 79: “*Partial oxidation induced* peak at 22.5° in the X-ray diffraction (XRD) spectrum indicates the formation of a stage-1 graphite intercalation compound with an interlayer distance of 8.0 \AA (Supplementary Fig. 6)^{27,28}.”

Question 4: *The reported graphene slurry has a high concentration of flocculating single-layer graphene sheets at pH of 14, and authors compared this mixture to GO solutions with well dispersed nano-sheets. A more direct comparison is between GO solution and graphene slurry at pH = 12, the stable pH for storing and utilizing graphene slurry.*

Response: We have supplied new control experiments on the GO solution and graphene slurry at pH = 12 in the revised version. Both GO and graphene dispersion at pH = 12 can be stored for weeks without obvious aggregations in Supplementary Figure 7 (a, b). The maximum dispersion concentration is similar to those reported in the literature ($\sim 0.1 \text{ mg mL}^{-1}$ for graphene in water), whereas our concentrated graphene slurry (pH = 14) can be re-dispersed on-demand. To further examine its stability for practical applications, we have also examined the TEM image and Zeta-potential of graphene dispersion at pH = 12 after storing for 1 week. As shown in Supplementary Figure 7 (c), the graphene flakes are loosely packed into few-layer morphology after solvent evaporation. We can observe many NaOH absorbates on graphene flakes in TEM image, which contributes to the surface potential against π - π re-stacking. Additionally, the Zeta potential of graphene dispersion after 1 week (-42.4 mV) is very close to the value of fresh solution (-48.4 mV), indicating excellent stability of graphene dispersion in this condition. Related discussions have been included in the Supplementary Information.

Supplementary Figure 7. (a,b) Digital photos of graphene and GO dispersion at pH = 12 after storing for a week, both at 0.1 mg mL⁻¹. (c) TEM image of graphene dispersion at pH = 12 after a week.

Question 5: *At line 253, the authors claim that sulfuric acid can be recovered and recycled. This may be easier said than done. Can the authors demonstrate this or provide relevant references to support this?*

Response: To address the reviewer's concern, we have supplied a video recording the recovery of sulfuric acid in the revised version (Supplementary Video 1). Details on the experimental setups can be found in Supplementary Information or our previous publication (Reactivity-controlled preparation of ultra-large graphene oxide by chemical expansion of graphite, *Chem. Mater.* **2017**, 29, 564-572):

Line 251: "After reaction, the black flakes were filtered through a 200 mesh sieve (Supplementary Figure 11 and Supplementary Video 1) and separated from dark green solution."

Supplementary Figure 11. (a) Mesh filtration of partially oxidized graphite to recover sulfuric acid; (b) Filter cake of the partially oxidized graphite.

Reviewer #3:

*The manuscript by Dong et al. proposes the new approach for the liquid exfoliation of graphite to yield graphene. To achieve better exfoliation, the authors subject graphite to partial oxidation with potassium permanganate is sulfuric acid, and then subject the as-obtained product to shear in aqueous solution of NaOH at pH=14. The difference from the previous studies is that the product is obtained not as stable dispersion, but in the form of flocculated aqueous slurry with concentrations as high as 5 wt%. The presence of adsorbed ions prevents the irreversible restacking of graphene flakes and enables their re-dispersion in solution on demand. The authors further demonstrate the use of the as-made graphene slurry for 3D printing. This approach is **novel, and has great potential for practical applications**, since avoiding copious amounts of water and/or organic solvents helps one to lower the manufacturing cost, and to minimize expenses with further storage and processing.*

The only concern I have relates to the experimental data, confirming the quality of as-obtained graphene.

Response: We thank the reviewer for his/her positive comments on our paper. We have addressed his/her concerns on the quality of as-obtained graphene, especially on the Raman spectra of graphene sheet in the revision.

Question 1: *On the first step authors use 1:1 weight ratio of graphite-to-KMnO₄. At this ratio, significant oxidation of graphite is normally registered. See for example: Carbon 1991, 29, 469-474; Carbon 1995, 33, 1585-1592; Chem. Mater. 2006, 18, 2740-2749; Carbon 2013, 53, 38-49; ACS Nano, 2014, 8, 3060-3068.*

As shown in the latter study, at this graphite/oxidizer ratio, roughly one half of the body of a graphite flake is converted to graphite oxide. The question: what authors do differently, that they obtain almost non-oxidized graphene? Shorter exposure time, lower temperature, etc.? Explanation for this apparent discrepancy with the literature data should be given.

Response: We appreciate the critical comments on graphite oxidation. In comparison with the listed (Ref 37-41) and related references, there're several reasons for the much lower oxidation degree in our case:

1) we use **graphite with bigger size** (100 mesh \approx 150 μ m), which can significantly reduce the diffusion and intercalation of sulfuric acid and oxidant into the interlayer spacing of graphite;

2) we use a **much lower oxidant to graphite ratio** (1 wt. equiv. of KMnO_4 for initial feeding and \sim 0.8 wt equiv. of KMnO_4 for final consumption) to avoid over-oxidation, while GO preparation usually requires over 3 wt. equiv. of KMnO_4 together with 0.5 \sim 1 wt. equiv. of NaNO_3 (for *in-situ* generation of HNO_3);

3) the **hydrolysis of graphite intercalated compounds** has been identified as a key step in graphite oxidation (*J. Am. Chem. Soc.* **2012**, *134*, 2815). Conventional GO preparation involves water injection into the reaction mixture (conc. H_2SO_4 etc.), this causes a sudden temperature increase which facilitate the oxidation of graphite.

In contrast, we **recover the excess sulfuric acids prior to hydrolysis** and pour the intercalated compounds into **a large amount of ice-water** (20 wt. equiv.) to minimize the heat generation from concentrated sulfuric acids during hydrolysis. We do not keep the hydrolysed compounds for further reaction. Excess amount of 30 wt% H_2O_2 is injected to terminate the reaction immediately;

4) the **reaction temperature** is low (r.t.) with a short reaction time (2 h).

A detailed comparison is presented in Table 1 and Supplementary Table 3. Related discussions have been added in this version.

It is also necessary to explain why we choose such experimental parameters for partial oxidation. As shown in Supplementary Figure 13, we examined the influence of KMnO_4 wt. equiv. on the final yield and quality of graphene. We found that 1 wt. equiv. of KMnO_4 are optimized for both yield and quality, whereas using a lower amount of KMnO_4 results in insufficient intercalation and a much worse graphene yield. Raman analysis reveals that for treatment conditions using low oxidant to graphite ratios (0.4 or 1.0 wt. equiv.), the aromatic structure of graphene is well-reserved with a low I_D/I_G ratio. Further increase in oxidant usage will dramatically reduce the quality of exfoliated graphene and finally convert graphene into graphene oxide. Based on the above observations, the optimized conditions (1 wt. equiv. KMnO_4 etc) were used in this manuscript to reduce unnecessary oxidations.

Supplementary Figure 13. (a) Graphene yields are greatly influenced by the ratio of oxidant to graphite, suggesting that appropriate oxidation is essential; (b) Raman spectra of graphite at various KMnO_4 ratios, together with a comparison of GO. Inset of (a), photo of graphite at different KMnO_4 ratio, (I) 0.4 wt. equiv., (II) 1 wt. equiv., (III) 2 wt. equiv. and (IV) 4 wt. equiv.

Table 1. Comparison between our approach and the oxidation exfoliation of graphite.

Ref	Method	Graphite	Pre-treatment	Ingredients per 1 g graphite	Temp.	Time	Hydrolysis
Ours	--	100 mesh	Mix in ice-bath	1 g KMnO ₄ , 20 mL H ₂ SO ₄	r. t.	1 h	Recover H ₂ SO ₄ ; pour into ice water
18	Hummer's	30 μm	Mix in ice-bath	3 g KMnO ₄ , 23 mL H ₂ SO ₄ , 0.5 g NaNO ₃	35 °C	0.5 h	Add 46 mL H ₂ O, react at 98 °C for 15 min
19	Hummer's	30 μm	Mix in ice-bath	3 g KMnO ₄ , 23 mL H ₂ SO ₄ , 0.5 g NaNO ₃	35 °C	0.5 h	
20	Modified Hummer's	70 μm	Mix in ice-bath	6 g KMnO ₄ , 46 mL H ₂ SO ₄ , 1 g NaNO ₃	35 °C	1 h	Add 80 mL H ₂ O, react at 90 °C for 30 min
21	Modified Hummer's	325 mesh	Mix at 90 °C	1 g K ₂ S ₂ O ₈ , 5 mL H ₂ SO ₄ , 1 g P ₂ O ₅	80 °C	4.5 h	Diluted with 2 L water
22	Modified Hummer's	70 μm	Mix in ice-bath	6 g KMnO ₄ , 46 mL H ₂ SO ₄ , 1 g NaNO ₃	35 °C	1 h	Add 80 mL H ₂ O, react at 90 °C for 30 min
24	Modified Hummer's	70 μm	Mix in ice-bath	3 g KMnO ₄ , 23 mL H ₂ SO ₄ , 0.5 g NaNO ₃	35 °C	0.5 h	Add 46 mL H ₂ O, react at 98 °C for 15 min
25	Modified Hummer's	300 μm	Mix in ice-bath	3 g KMnO ₄ , 23 mL H ₂ SO ₄ , 0.5 g NaNO ₃	35 °C	0.5 h	
26	Modified Hummer's	--	Mix in ice-bath	3.5 g KMnO ₄ , 25 mL H ₂ SO ₄	35 °C	2 h	Excess H ₂ O at 0 °C
27	Modified Hummer's	--	Mix in ice-bath	6 g KMnO ₄ , 46 mL H ₂ SO ₄ , 1 g NaNO ₃	35 °C	1 h	Add 80 mL H ₂ O, react at 90 °C for 30 min
28	Modified Hummer's	--	Mix in ice-bath	3.5 g KMnO ₄ , 25 mL H ₂ SO ₄	35 °C	2 h	Excess H ₂ O at 0 °C
29	Modified Hummer's	30 μm	Mix in ice-bath	3 g KMnO ₄ , 23 mL H ₂ SO ₄ , 0.5 g NaNO ₃	35 °C	0.5 h	Add 46 mL H ₂ O, react at 98 °C for 15 min
30	Modified Hummer's	30 μm	Mix in ice-bath	3 g KMnO ₄ , 23 mL H ₂ SO ₄ , 0.5 g NaNO ₃	35 °C	0.5 h	
37	Modified Hummer's	10 μm	Mix in ice-bath	3 g KMnO ₄ , 23 mL H ₂ SO ₄ , 0.5 g NaNO ₃	35 °C	0.5 h	
38	Staudenmaier's	40 - 47μm	Mix in ice-bath	11 g KClO ₃ , 17.5 mL H ₂ SO ₄ , 9 mL fuming HNO ₃	0 °C	24 - 240 h	Pour into 1 L water
39	Brodie's	250 - 500 μm	Mix in ice-bath	8.5 g NaClO ₃ , 6 mL fuming HNO ₃	60 °C	8 h	Pour into 0.1 L water

40	Modified Hummer's	20 μm	Mix at $< 20\text{ }^\circ\text{C}$	0.5 - 3 g KMnO_4 , 22.5 mL H_2SO_4	35 $^\circ\text{C}$	2 h	Add 45 mL H_2O
41	Modified Hummer's	100 mesh	Mix at r.t.	1 - 4 g KMnO_4 , 150 mL H_2SO_4	r.t.	hours	N. A.
42	K_2FeO_4 -based	40 μm	N.A.	6 g K_2FeO_4 , 40 mL H_2SO_4	r.t.	1 h	Wash by H_2O
43	Staudenmaier's	45 μm	Mix in ice-bath	11 g KClO_3 , 17.5 mL H_2SO_4 , 9 mL fuming HNO_3	r.t.	96 h	Add excess H_2O
44	Modified Hummer's	150 μm	Mix at 35 $^\circ\text{C}$	6 g KMnO_4 , 120 mL H_2SO_4 , 13.3 mL H_3PO_4	50 $^\circ\text{C}$	12 h	Pour onto ice
45	Staudenmaier's	--	Mix in ice-bath	11 g KClO_3 , 17.5 mL H_2SO_4 , 95 mL fuming HNO_3	r.t.	96 h	Pour into 0.8 L H_2O
46	Modified Hummer's	--	Ground with NaCl	3 g KMnO_4 , 23 mL H_2SO_4 , 0.1 g NaNO_3	70 $^\circ\text{C}$	24 h	Wash by H_2O
47	Modified Hummer's	24 μm	Mix in ice-bath	4.5 g KMnO_4 , 62.1 g H_2SO_4 , 0.75 g NaNO_3	r.t.	120 h	Add 46 mL H_2O , react at 98 $^\circ\text{C}$ for 15 min
48	Modified Hummer's	3 - 5 mm	N.A.	7.5 g KMnO_4 , 66 mL H_2SO_4	80 $^\circ\text{C}$	8.5 h	Sonicate in water bath for 1 - 6 h
49	Modified Hummer's	--	Mix in ice-bath	6 g KMnO_4 , 23 mL H_2SO_4 , 50 g NaCl	105 $^\circ\text{C}$	14.5 h	Add 46 mL H_2O , react at 98 $^\circ\text{C}$ for 15 min
50	Brodie's	74 μm	Mix in ice-bath	8.5 g NaClO_3 , 20 mL fuming HNO_3	--	24 h	Pour into 0.1 L water
51	Hummer's	45 μm	Pre-oxidation by $\text{K}_2\text{S}_2\text{O}_8$ & P_2O_5	8.75 g KMnO_4 , 82.5 mL H_2SO_4	80 $^\circ\text{C}$	10 h	Dilute with 250 mL H_2O at 50 $^\circ\text{C}$
52	Hummer's	350 μm		400 g KMnO_4 , 867 mL H_2SO_4	80 $^\circ\text{C}$	9.5 h	Stir with H_2O for 2 h
53	Modified Hummer's	--	Ground with NaCl	0.75 g KMnO_4 , 23 mL H_2SO_4 , 0.1 g NaNO_3	40 $^\circ\text{C}$	25h	Add 46 mL H_2O , react for 25 min

References:

37. Mermoux, M., Chabre, Y. & Rousseau, A. FTIR and ^{13}C NMR study of graphite oxide. *Carbon* **29**, 469-474 (1991);
38. Hontoria-Lucas, C., López-Peinado, A., López-González, J. de D., Rojas-Cervantes, M. L. & Martín-Aranda, R. M. Study of oxygen-containing groups in a series of graphite oxides: physical and chemical characterization. *Carbon* **33**, 1585-1592 (1995);
39. Szabó, T. *et al.* Evolution of surface functional groups in a series of progressively oxidized graphite oxides. *Chem. Mater.* **18**, 2740-2749 (2006);

40. Krishnamoorthy, K., Veerapandian, M., Yun, K. & Kim, S.-J. The chemical and structural analysis of graphene oxide with different degrees of oxidation. *Carbon*, **53**, 38-49 (2013);
41. Dimiev, A. M. & Tour, J. M. Mechanism of graphene oxide formation. *ACS Nano* **8**, 3060-3068 (2014);
42. Peng, L. *et al.* An iron-based green approach to 1-h production of single-layer graphene oxide. *Nat. Commun.* **6**, 5716 (2015);
43. McAllister, M. J. *et al.* Single sheet functionalized graphene by oxidation and thermal expansion of graphite. *Chem. Mater.* **19**, 4396-4404 (2007);
44. Marcano, D. C. *et al.* Improved synthesis of graphene oxide. *ACS Nano* **4**, 4806-4814 (2010);
45. Lomeda, J. R., Doyle, C. D., Kosynkin, D. V., Hwang, W. F. & Tour, J. M. Diazonium functionalization of surfactant-wrapped chemically converted graphene sheets. *J. Am. Chem. Soc.* **130**, 16201-16206 (2008);
46. Wang, H., Robinson, J. T., Li, X. & Dai, H. Solvothermal reduction of chemically exfoliated graphene sheets. *J. Am. Chem. Soc.* **131**, 9910-9911 (2009);
47. Mkhoyan, K. A. *et al.* Atomic and electronic structure of graphene-oxide. *Nano Lett.* **9**, 1058-1063 (2009);
48. Su, C. *et al.* Electrical and spectroscopic characterizations of ultra-large reduced graphene oxide monolayers. *Chem. Mater.* **21**, 5674-5680 (2009);
49. Liu, Z., Robinson, J. T., Sun, X. & Dai, H. Pegylated nanographene oxide for delivery of water-insoluble cancer drugs. *J. Am. Chem. Soc.* **130**, 10876-10877 (2008);
50. Shin, H. *et al.* Efficient reduction of graphite oxide by sodium borohydride and its effect on electrical conductance. *Adv. Funct. Mater.* **19**, 1987-1992 (2009);
51. Tang, L. *et al.* Preparation, structure, and electrochemical properties of reduced graphene sheet films. *Adv. Funct. Mater.* **19**, 2782-2789 (2009);
52. Li, X. *et al.* Simultaneous nitrogen doping and reduction of graphene oxide. *J. Am. Chem. Soc.* **131**, 15939-15944 (2009);
53. Luo, Z., Lu, Y., Somers, L. A. & Johnson, A. T. C. High yield preparation of macroscopic graphene oxide membranes. *J. Am. Chem. Soc.* **131**, 898-899 (2009).

Supplementary Table 3. Comparison between our approach and chemical reduction of graphene oxide.

Ref	Post Reduction	Temp. (°C)	Time (h)	C/O	Thickness	I _D /I _G	Conductivity (S m ⁻¹)
Our work	N. A.	--	--	16.0	Single layer, 90%	0.22 ~ 0.33	25,200 (42,400 after HI-reduction)
18	Hydrazine	100	24	10.3	-	> 1	200
21	Hydrazine	r. t.	~168	-	-	>1	-
22	HI + acetic acid	40	40	15.27	-	>1	30,400
19	HI	100	1	12	-	>1	29,800
20	Na + NH ₃	-78	0.5	16.6	-	>1	350 Ω, 80% transmittance
23	Na + ethanol	220	72	6.4	Single layer	1.16	0.05
24	NaBH ₄	80	1	4.8	-	1.9	82
25	Fe + HCl	r. t.	6	7.9	2-10 layers	0.32	2,300
26	Propylene carbonate	150	12	8.3	-	-	2,100
27	H ₂ O	180	6	5.6	Single layer	0.9	-
28	Hydrazine + DMF/H ₂ O	80	1	11	Single layer	>1	1,700; 16,000 when dried at 150 °C
29	H ₂ O	95	48	6	Single layer, 65%	-	-
30	Argon	500	1	-	-	-	35,100

Question 2: *The Raman mapping (Fig. 2b) does not match the rest of the provided experimental data. The AFM and TEM images show single-layer graphene. The Raman spectrum does not represent the single-layer graphene flake. This is apparent from the G/2D ratio and the character of the D and G-bands. The Raman spectrum of a single-layered graphene is very different: 2D signal must be higher than G-band. Here the situation is opposite, strongly attributing the given spectrum to multi-layered graphene (or graphite) with number of layers >5. Both D-band and G-band are broadened, suggesting highly damaged structure, with the density of defects similar to that in GO (this is logical in light of the oxidative treatment). Thus, the Raman spectrum represents the multi-layered graphene flake, where the top layer is fully oxidized, i.e. the top layer is graphene oxide. It is very much the same GNP as the product obtained in ref. [25], authors citing. The question: if the authors obtain mostly single-layered graphene, as they show on the AFM images (Fig. 2,e; Fig. S2; Fig. S3), then why they picked the multi-layered oxidized flake for the Raman mapping? All the data should be in accordance with each other.*

Response: In order to collect AFM and TEM data, we spin/drop-coated single-layer sheets aqueous dispersion on mica substrate/TEM grid and hence the data is representative of monolayer flake. Our Raman was collected from the dried, aggregated samples. We have re-dispersed the samples in an attempt to collect better quality Raman that is representative of monolayers, however, the best data in Figure 2 show an I_{2D}/I_G ratio of ~ 0.45 , corresponding to \sim three-layer graphene (Raman spectroscopy in graphene, *Phys. Rep.*, **2009**, 473, 51-87; Raman spectrum of graphene and graphene layers, *Phys. Rev. Lett.*, **2006**, 97, 187401). The data we have obtained still lead us to conclude that what we have is predominantly single layer graphene because of the following reasons:

1) We used ***Si wafer*** for Raman measurements since the transparent mica shows low optical contrast and cannot be used for sample tracing. Due to the different wettability of Si wafer and mica, we have to use ***NMP*** to re-disperse and spin-coat graphene onto Si wafer. This leads to a huge problem in getting isolated graphene sheets because of the high-boiling point of NMP and serious coffee-ring effect (Bi- and trilayer graphene solutions, *Nat. Nanotech.* **2011**, 6, 439-444). Most graphene sheets are stacked with each other.

2) It is well-known that ***defects*** have a strong influence on the Raman spectrum of graphene. Although the 2D band is not so sensitive to the defective sites as the D band, it is possible that the intensity of the 2D band becomes reduced after oxidation, functionalization or other treatments (Spectroscopy of covalently functionalized graphene, *Nano Lett.* **2010**, 10, 4061-4066). In fact, for ***solution exfoliated graphenes***, it is generally difficult to obtain a similar I_{2D}/I_G ratio as defect-free, mechanically exfoliated graphenes. A detailed comparison between our Raman data and the literature is provided below, from which we can find ***our Raman data provide a relatively high I_{2D}/I_G ratio*** for the solution exfoliated graphenes.

Figure 2. (a-c) Raman spectra of graphite, partially oxidized graphite and graphene sheets.

In this regard, although we are not able to provide the Raman spectrum mentioned by the Reviewer, we believe it is a technical issue related to the processing of Raman measurements. To clarify this mismatch between Raman and AFM/TEM, we have revised the Main Text Figure 2b and added a short comments in the revision:

Line 111: “The presence of the 2D band at $\sim 2,700\text{ cm}^{-1}$ reflects the well-preserved aromatic structure of graphene, which is absent or negligible in reduced graphene oxide (rGO). The I_{2D}/I_G ratio is ~ 0.45 , corresponding to that of ~ 3 layer graphene³³. Since we have to spin-coat NMP dispersion on Si wafer for Raman, it is difficult to avoid the re-stacking of graphene nanosheets during solvent evaporation.”

Main Text Figure 2. Quality of exfoliated graphene flakes. (a) Atomic resolution STEM image of a graphene flake. Inset shows the corresponding magnified image with perfect graphene lattice. (b) Raman spectrum of graphene, showing I_D/I_G of 0.23. (c) XPS C_{1s} spectrum of graphene. (d-f) Wide-field AFM image of graphene flakes and the corresponding thickness (e) and lateral size histograms (f).

Figure 3. Raman spectra of solution exfoliated graphene from the literature.

Question 3: Based on Fig. 2b, I would expect the flakes to be thicker along the perimeter. The AFM height profiles for GO normally show 0.9-1.2 nm height. Graphene - 0.5-0.6 nm. Could the authors, please, comment on this in the manuscript text? If the flakes are indeed thicker on the edges, as I can see from the provided height-profiles for some flakes, discussing this in the text will strengthen the publication. I would recommend to acquire a high resolution image of one single flake and carefully examine it.

Response: According to the reviewer's suggestion, we have provided a high resolution AFM image in Figure 4 to examine the thickness at the edge and the basal plane. Basically, we **do not observe such a different thickness** in our sample, which can be explained by the following reasons:

1) We used **tapping AFM** to acquire the images and fluctuation in edge thickness is due to instrumental error and can be corrected by slowing down the scan speed of the AFM. The AFM line scan in Main Text Figure 2d is collected relatively fast and over a large scan area (~ 25 μm), resulting in a large instrument error (lower thickness at basal plane). However, when we performed a high resolution AFM scan, such fluctuation at the edge is quite small for our graphene (Supplementary Figure 2), indicating that the thickness at the edge is almost similar to that of the basal plane;

2) It is a well-known phenomenon that **water (or small molecules) is trapped between graphene and substrate**, resulting in an apparent higher thickness (e.g., 0.9 - 1.2 nm for graphene oxide) compared to the monolayer thickness of graphene (0.34 nm). Partial intercalation at the edge is difficult due to the large local strain. Thus, we observed similar thickness (0.5 ~ 0.6 nm) at the edge and basal plane, we do not think we need to discuss this at length.

Supplementary Figure 2. AFM images of exfoliated graphene sheets. Scale bar, 2 μm .

Figure 4. High resolution AFM image of exfoliated graphene sheet.

Question 4: *The SEM images of as-made graphene (Fig. 4b and Fig. S1c) also look more like those for multi-layered GNP. Thus, the two data (Raman and SEM) provided by the authors, suggest that they obtain GNP, but not single-layered graphene, as they claim. The 3D printing and the fabricated aerogels do not confirm the single-layer character.*

The complete exfoliation of graphite source to the single-layer graphene is the main claim of this study that makes it, according to the authors, advantageous over the previous works. Thus, this needs to be unambiguously confirmed.

Response: We thank the reviewer for the critical comments. Main Text Figure 4b and Supplementary Figure 1 (c) in the original manuscript were obtained from freeze-dried graphene powders, where aggregation was unavoidable. We have supplied new SEM image by spin-coating graphene NMP dispersion onto Si substrate to minimize such aggregation. We also compare our SEM images with the literature (Figure 5), where our graphene nanosheet has a similar morphology to the “single-layer” graphene. Owing to the coffee-ring effect during solvent evaporation (*Nat. Nanotechnol.* **2011**, *6*, 439; *J. Phys. Chem. C* **2014**, *118*, 27081), it is a natural phenomenon that re-stacking of graphene nanosheets occurs. Thus, it is challenging to determine the single-layer morphology solely from the SEM images.

We employed two additional techniques: AFM and TEM/SAED, to validate that our graphene nanosheets are single-layer. As shown in Figure 2 (d) in manuscript, the height profile together with a statistical analysis of over 100 flakes show that > 90 % of the flakes are single layer (<1 nm in thickness) in AFM. The characteristic SAED pattern and new high-resolution TEM images in Supplementary Figure 3 and 4 also confirm the single-layer morphology, although we can occasionally find some two or three-layer graphene (Supplementary Figure 4 (h), (i)). To explain the mismatch between Raman and AFM, we also supply new Raman spectra in the revised version. Please refer to our response to your Questions 2, 3.

Supplementary Figure 1d. SEM image of graphene sheets on Si wafer.

Supplementary Figure 3. TEM image and the corresponding SAED pattern of single-layered graphene sheet.

Supplementary Figure 4. (a-i) HRTEM images of graphene sheets. Scale bar, 5 nm.

Figure. 5. SEM images of graphene derivatives from the literature. (a-d) graphene nanosheets; (e) reduced graphene oxide; (f) chemical converted graphene (g-i) graphene oxide nanosheets.

Question 5: *I will be happy to support after addressing these questions. Major revision is suggested.*

Response: Thank you for the critical comments. We believe we have addressed your questions in the revision.

REVIEWERS' COMMENTS:

Reviewer #2 (Remarks to the Author):

The authors have made quite significant efforts to address reviewer's concerns. All of the concerns raised in my previous review have been adequately resolved. The new result presented in supplementary Figure 7 are a bit confusing. Figure S7b shows a vial of GO dispersion under pH 12 at 0.1 mg/ml. At this concentration and pH, GO would be partially reduced and should appear a lot darker. I would suggest the authors double check this panel to make sure they did not inadvertently use the wrong photo.

Either way, I do not need to see another revision again. The editor should feel sufficiently confident to make an editorial decision.

Reviewer #3 (Remarks to the Author):

The authors addressed my questions in full. The manuscript can be published as is.

Point-by-point responses to comments for manuscript NCOMMS-17-20613A

“A Non-Dispersion Strategy for Large-Scale Production of Ultra-High Concentration Graphene Slurries in Water”

Reviewer #2:

“The authors have made quite significant efforts to address reviewer's concerns. All of the concerns raised in my previous review have been adequately resolved. The new result presented in supplementary Figure 7 are a bit confusing. Figure S7b shows a vial of GO dispersion under pH 12 at 0.1 mg/ml. At this concentration and pH, GO would be partially reduced and should appear a lot darker. I would suggest the authors double check this panel to make sure they did not inadvertently use the wrong photo.

Either way, I do not need to see another revision again. The editor should feel sufficiently confident to make an editorial decision.”

We thank the Reviewer for his/her positive assessment. We have replaced the Supplementary Fig. S7b with the following image:

Supplementary Figure 7. (a,b) Digital photos of graphene and GO dispersion at pH = 12 after storing for 1 week, both at 0.1 mg mL⁻¹. (c) TEM image of graphene dispersion at pH = 12 after 1 week. Scale bar: c. 5 μm.

Reviewer #3:

“The authors addressed my questions in full. The manuscript can be published as is.”

We thank the Reviewer for his/her positive assessment.